# Static and Fatigue Tensile Properties of Cross-Ply Carbon-Fiber-Reinforced Epoxy-Matrix-Composite Laminates with Thin Plies

**Kimiyoshi Naito** [1,2,*], **Yuto Seki** [1,3,4] **and Ryo Inoue** [3]

1. Polymer Matrix Composites Group, Research Center for Structural Materials, National Institute for Materials Science (NIMS), Tsukuba 305-0047, Japan
2. Department of Aerospace Engineering, Tohoku University, Sendai 980-8577, Japan
3. Department of Mechanical Engineering, Tokyo University of Science, Tokyo 162-8601, Japan; inoue.ryo@rs.tus.ac.jp
4. Aero Engine, Space and Defense Fields, IHI Corporation, Tokyo 135-8710, Japan
* Correspondence: naito.kimiyoshi@nims.go.jp; Tel.: +81-2-9859-2803

**Abstract:** Carbon-fiber-reinforced epoxy-matrix composite (CFRP) laminates with thin plies have strong damage-resistance properties compared with standard prepregs. The static and fatigue tensile fracture behavior of cross-ply CFRP laminates with thin plies should be further studied to establish the applicability of thin-ply prepregs for industrial structures. In this study, the static and fatigue tensile properties of cross-ply, high-strength polyacrylonitrile (PAN)-based carbon-fiber (T800SC)-reinforced epoxy-matrix composites with thin plies were investigated. The fiber orientations of the CFRP specimens were set to cross-ply with $[0/90]_{10S}$ (subscript S means symmetry), $[(0)_5/(90)_5]_{2S}$, and $[(0)_{10}/(90)_{10}]_S$. The static and fatigue tensile characteristics of the cross-ply CFRPs with thick plies with $[0/90]_{2S}$ and $[(0)_2/(90)_2]_S$ were also investigated for comparison. Under static loading, the tensile strength and failure strain of the thinnest 90°-ply-CFRP specimens were more than 5% higher than those of the other 90°-ply-thickness specimens. However, the tensile moduli and Poisson's ratios were comparable between the cross-ply CFRPs with thin and thick plies. Under fatigue loading, the fatigue responses of the thinnest 90°-ply-CFRP specimens were 3% higher than those of the other 90°-ply-thickness specimens during lower-fatigue-cycle testing (<$10^5$ cycles). However, during higher-fatigue-cycle testing (>$10^5$ cycles), the fatigue responses decreased, with a decrease in the 90°-ply thickness, and the fatigue characteristics of the thinnest 90°-ply-CFRP specimen were 7% lower than those of the other cross-ply thin- and thick-ply-CFRP specimens.

**Keywords:** CFRP; cross-ply; ply thickness; static and fatigue properties

---

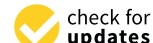



## 1. Introduction

Carbon-fiber-reinforced polymer-matrix composites (CFRPs) have gained attention from the aerospace, automotive, and sporting-goods industries [1–5]. Carbon-fiber-reinforced polymer-matrix-composite-laminates are affected by the buildup of damage (e.g., matrix cracking, delamination), and strong CFRPs can be developed to achieve an increase in damage resistance [6–10]. The fracture behavior in CFRP laminates includes matrix cracking, which can take the form of delamination, fiber–matrix debonding, and fiber fracture and splitting. Transverse microcracking related to the thickness of the ply occurs, and delamination damage follows. Many theoretical and practical studies have been performed on CFRP laminates' fracture behavior, including delamination induced by transverse cracks [11–18]. Laws and Dvorak reported transverse cracking in cross-ply-composite laminates based on statistical fracture mechanics. The crack density as a function of the applied load was also assessed [11]. Lim and Hong investigated the effect of transverse cracks on the thermomechanical-property degradation of cross-ply-composite laminates [12]. Lee

and Daniel proposed a simplified shear-lag analysis using a progressive damage scheme for cross-ply-composite laminates under uniaxial tensile loading. Closed-form solutions for stress distributions, the transverse crack density, and the reduced stiffnesses of damaged plies were obtained as a function of the applied load [13]. Takeda and Ogihara evaluated microscopic fracture mechanisms, focusing on the transverse cracks in CFRP cross-ply laminates [14–16].

Ply thickness significantly affects the increase in damage accumulation, and thinner plies have higher transverse strength or resistance to matrix cracking [6–10,19–21]. Furthermore, using thin-ply laminates with fewer ply angles makes laminate homogenization much easier, leading to many benefits for laminate design and optimization. Thin-ply materials are particularly suitable for application in many components subjected to combined bending and twisting loads and deformations, such as wings, stabilizers, turbine fans, wind turbines, helicopter rotor blades, propellers and shafts, torsional rods, and various sporting goods [22,23]. Some experimental studies and a failure analysis using a damage model of thin-walled composite structures were reported in the literature [24–28]. Sasayama et al. [19] developed thin-ply prepregs using a novel technique involving the spreading of carbon-fiber tows. Carbon-fiber-reinforced polymer-matrix composites laminates with thin plies have stronger damage-resistance properties than standard prepregs [19–21].

Nevertheless, there are still few studies on CFRP laminates with thin plies, and there are no studies investigating the relationship between ply thickness and the static and fatigue tensile properties of CFRP laminates with thin plies with various-stacking sequences. The static and fatigue tensile fracture behavior of cross-ply CFRP laminates with thin plies should be further studied to establish the applicability of thin-ply prepregs for industrial structures.

Cross-ply laminates have been extensively investigated via theoretical and experimental studies because this is a basic laminate configuration [12–16]. In this work, static and fatigue tensile tests of cross-ply, high-strength polyacrylonitrile (PAN)-based carbon-fiber (T800SC)-reinforced epoxy-matrix-composite laminates with thin plies with three stacking sequences ($[0/90]_{10S}$, $[(0)_5/(90)_5]_{2S}$, and $[(0)_{10}/(90)_{10}]_S$.) were conducted to examine their fracture behavior. In situ static and fatigue tensile tests using a digital microscope were performed to obtain practical knowledge of the fracture mechanism of the tensile properties. The relationships between the crack density, number of cracks, and tensile properties were examined according to the damage observed. In addition, an analytical model using the cumulative damage to the composite was applied to estimate the fatigue lifetimes.

## 2. Materials and Methods

### 2.1. Materials

A promising and cost-effective method is to use a tow-spreading technology developed by the Industrial Technology Center in Fukui Prefecture [19]. The CFRP laminates were produced using an epoxy-matrix-based thin-ply unidirectional (UD) prepreg (fiber: T800SC-24000-10E, matrix: 180 °C-cured-type epoxy). The T800SC carbon fiber is a high-strength PAN-based carbon fiber (Toray Industries, Inc., Tokyo, Japan). The Industrial Technology Center of Fukui Prefecture, Fukui, Japan supplied thin-ply UD prepreg. Thin-ply UD prepreg with nominal thicknesses of 0.032 mm (fiber-area weight (FAW): 30 g/m$^2$, resin content (RC): 38%) was used. Thick-ply UD prepreg (fiber: T800SC, matrix: 180 °C-cured-type epoxies), which Industrial Technology Center of Fukui Prefecture also supplied, with nominal thicknesses of 0.160 mm (FAW: 150 g/m$^2$, RC: 38%) was also used for comparison.

### 2.2. Specimen Preparation

The prepreg sheets were cut into suitable sizes and fiber orientations. The sheets were placed on a vacuum-molding board. The CFRP laminates were made using a hand lay-up and vacuum-bagging method (no bleeder). The fiber orientations of the CFRP laminates are shown in Figure 1. Cross-ply with thin ply with $[0/90]_{10S}$ (subscript 10 and S mean

repeating and symmetry), $[(0)_5/(90)_5]_{2S}$, and $[(0)_{10}/(90)_{10}]_S$ were also fabricated. Cross-ply CFRPs with thick ply with $[0/90]_{2S}$ and $[(0)_2/(90)_2]_S$ were also fabricated for comparison. The cross-ply CFRPs with thin plies ($[0/90]_{10S}$, $[(0)_5/(90)_5]_{2S}$, and $[(0)_{10}/(90)_{10}]_S$) and cross-ply CFRPs with thick plies ($[0/90]_{2S}$ and $[(0)_2/(90)_2]_S$) were designated as NA, NB, NC, WB, and WC, respectively. The fiber-volume fraction of the cross-ply CFRPs with thin and thick plies was 52.3%. The prepreg sheets were pressed at 490 kPa and cured at 180 °C for 4 h (the heating rate was 1 °C/min) using an autoclave (ACA Series, Ashida Mfg. Co., Ltd., Osaka, Japan) in the laboratory. The nominal thickness of the CFRP laminates was about 1.28 mm.

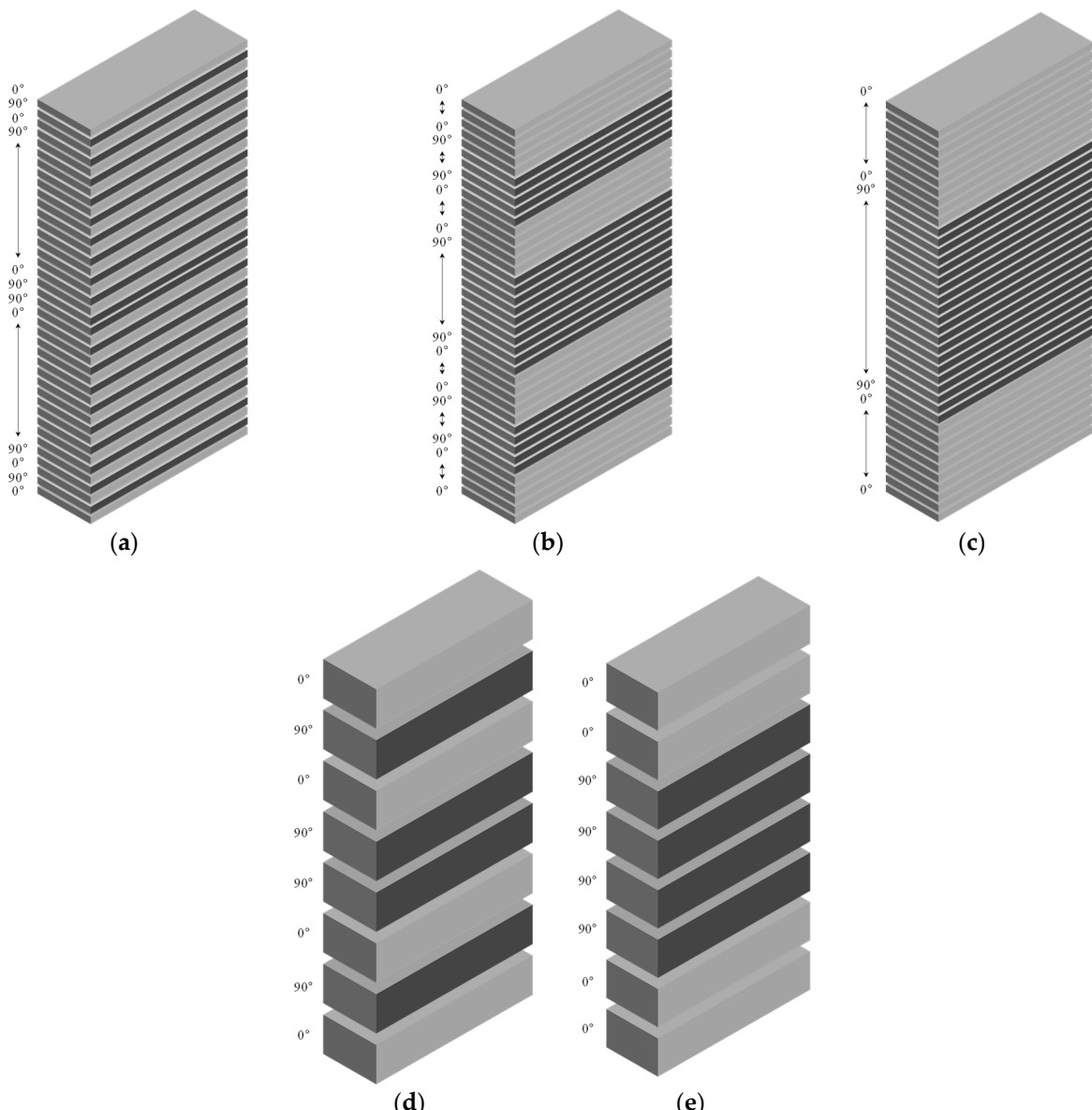

**Figure 1.** Schematic views of stacking sequences of the cross-ply CFRP with thin- and thick-ply laminates: (**a**) NA; (**b**) NB; (**c**) NC: (**d**) WB; (**e**) WC.

The fabricated CFRP laminates were cut into 10-mm × 10-mm pieces using a rotary cutting machine (Refine cutter RCA-234, Refine Tec Ltd., Hong Kong, China) at 2500 rpm with an abrasive cutting wheel (GC150NB, Heiwa Technica Co., Ltd., Tokyo, Japan). The CFRP samples were embedded in epoxy resin and subsequently polished by an automatic polishing machine (Automet 2000, Buhler Ltd., Kanazawa, Japan) with polycrystalline diamond suspensions of 6 μm and 3 μm and alumina suspension of 0.05 μm to produce

cross-sections of the CFRP samples for morphological observation. The cross-sectional morphology of the CFRP samples was also observed using a digital microscope (dual objective zoom lens (20× to 2000×), VH-ZST, in VHX-6000, Keyence, Osaka, Japan).

The CFRP laminates were cut into rectangular straight-side tensile-test specimens 200 mm long (gage length, *L*, of 100 mm) and 10 mm wide. Plain-woven-fabric glass-fiber-reinforced plastic (50 mm in length, 10 mm in width, and 1 mm in thickness) tapered tabs were affixed to the tensile-test specimen to minimize damage from the grips on the tensile testing machine. The fiber axis in the sample was oriented in line with the length of the tensile-test specimen (outer 0°-direction specimen). The cut edges of the tensile-test specimens were polished to eliminate the effects of stress concentrations caused by surface roughness. Similar specimen-preparation procedures of other composites have been described in the literature [4,29,30].

### 2.3. Static Tests

Static tensile tests of CFRP specimens were conducted using a universal testing machine (Autograph AG-series, Shimadzu Corp., Kyoto, Japan) with a load cell of 50 kN. The specimen was set up in the testing machine. A crosshead speed of 1.0 mm/min was applied, and all tests were performed in the laboratory environment at room temperature (at 23 °C ± 3 °C and 50% ± 5% relative humidity). The static tensile test gives a load (*P*) as a function of the extension (*U*$^*$) curve up to failure. The tensile stress ($\sigma_L$) and tensile strain ($\varepsilon_L$) were calculated as follows:

$$\sigma_L = \frac{P}{S}, \tag{1}$$

$$\varepsilon_L = \frac{U^*}{L^*}, \tag{2}$$

$$\nu_{LT} = -\frac{\varepsilon_{T(gauge)}}{\varepsilon_{L(gauge)}}, \tag{3}$$

where *S* is the total cross-sectional area of the CFRP specimens, which can be computed from the width and thickness, as determined using a micrometer. The *L*$^*$ is the distance between targets (reference marks). The targets were marked on the specimens (*L*$^* \approx 50$ mm). The extension (*U*$^*$) was measured using a non-contact video extensometer (DVE-201, Shimadzu Corp., Kyoto, Japan). The DVE-201 extensometers perform precise, non-contact elongation measurements using CCD cameras to capture digital images of test specimens. The longitudinal (tensile) strain ($\varepsilon_{L(gauge)}$) and transverse strain ($\varepsilon_{T(gauge)}$) were also determined using strain gauges. Similar static-test procedures of other composites have been applied in the literature [4,29,30]. Five specimens were tested for each instance according to ASTM D3039.

### 2.4. Fatigue Tests

Uniaxial fatigue tests of CFRP specimens were performed under sinusoidal waveform loading by a servo-hydraulic testing machine (MTS Landmark, MTS, Tokyo, Japan) with a 50-kN load cell at a frequency of 10 Hz. The stress ratio of the minimum stress to the maximum stress was 0.1. The maximum applied stresses were set as follows: 0.9, 1, 1.05, 1.1, 1.196, and 1.358 GPa for NA; 1.015, 1.05, and 1.302 GPa for NB; 1.05, 1.205, 1.356, and 1.375 GPa for NC; 1, 1.054, 1.1, 1.102, 1.131, and 1.2 for WB; 1, 1.003, 1.05, 1.075, 1.1, and 1.221 GPa for WC. One specimen was tested for each stress state to obtain the relation between the applied maximum stress, $\sigma_{max}$, and the number of cycles to failure, $N_f$, also defined as the *S*–*N* curves. The fatigue tests were terminated after $1 \times 10^7$ cycles. All tests were performed in a laboratory environment at room temperature (at 23 °C ± 3 °C and 50% ± 5% relative humidity). Similar fatigue-test procedures for other composites have been described in the literature [5,31–33].

### 2.5. In Situ Static and Fatigue Tests

Figure 2 shows the experimental set-up for in situ static and fatigue tensile tests. In situ static and fatigue tensile tests using a digital microscope (long-focal-distance, high-performance-zoom lens ($50\times$–$500\times$) VH-Z50L in VHX-6000, Keyence, Osaka, Japan) with a XYZ stage were performed using a servo-hydraulic testing machine (MTS Landmark, MTS, Tokyo, Japan) with a 50-kN load cell. The CFRP specimens used for in situ observation were directly polished by an automatic polishing machine using a specimen-fixing tool.

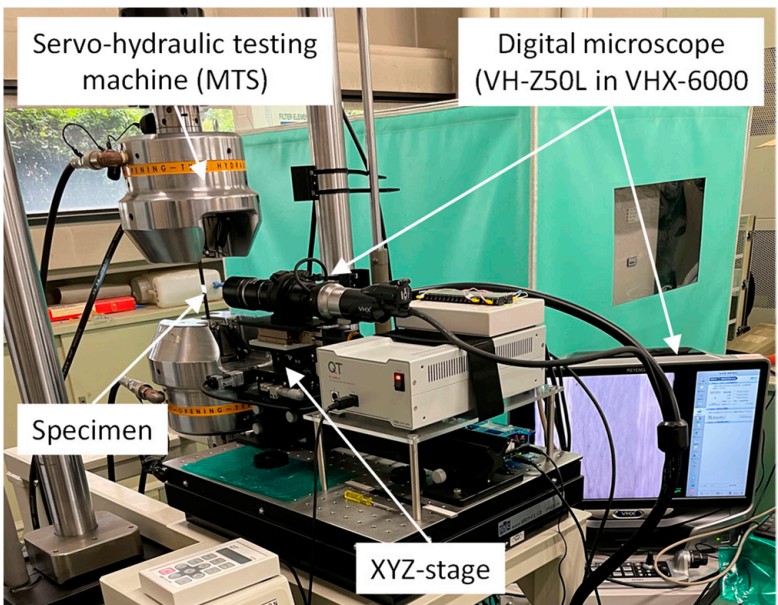

**Figure 2.** Experimental set-up for in situ static and fatigue tensile tests.

An automatic procedure was used for observation. The testing machine was paused when the loads or cycles reached a setting value. The XYZ stages were moved automatically to obtain the photos. Following the observation, the XYZ stage was also moved to safety, and the loads or fatigue cycles were restarted. The observation tests were repeated until the setting-end loads or cycles.

For the static tensile tests, the displacement after a predetermined load (100-MPa increment before 600 MPa and 50-MPa increment after 600 MPa) was stopped to allow the in situ digital-microscope observations.

Two testing procedures were applied for the fatigue tensile test, and the observation cycles were set to $10^4$. The aim of the first procedure (1) was to obtain the transverse-crack-onset stress under fatigue loading at $10^5$ cycles. The aim of the second (2) was to obtain the fatigue-damage-propagation behavior at same applied maximum stress. In method (1), a low cyclic stress (95 MPa) was applied to $10^5$ cycles. When the transverse crack was not observed, the cyclic stress increased by 32 MPa, and then the cyclic stress was applied to $10^5$ cycles. This procedure was used until a transverse crack was observed. In method (2), applied maximum stress was selected as 1.05 GPa (all specimens failed between $10^6$ and $10^7$ cycles) for each specimen.

## 3. Results

### 3.1. Morphologies

Figure 3 shows the digital-microscope images of through-thickness-edge-sectional view for the cross-ply CFRP samples with thin and thick plies. Each layer was observed, and the CFRP samples presented matrix-rich regions in the gaps between the layers. There were no noticeable voids in any of the samples of the cross-ply CFRP with thin and thick plies.

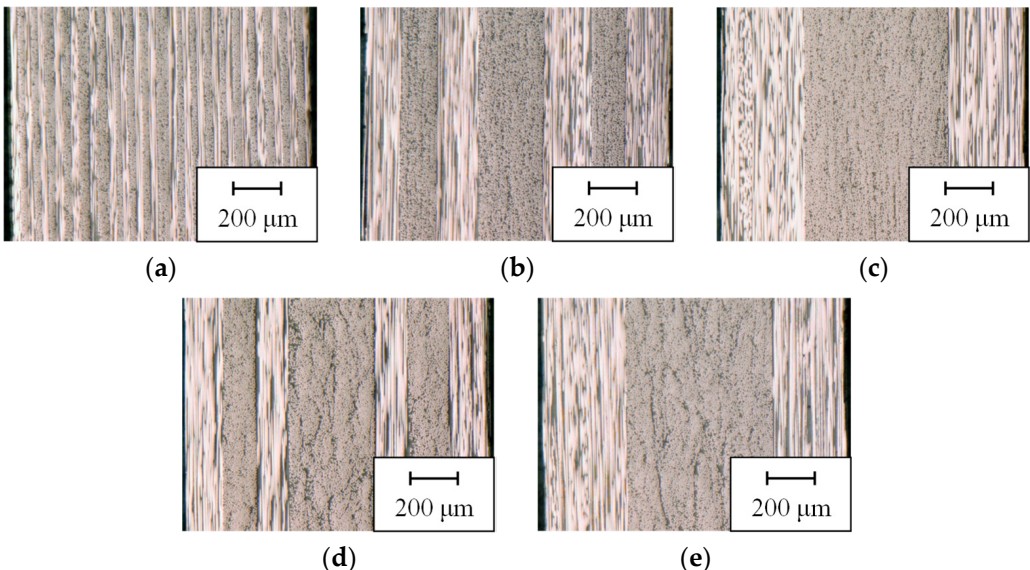

**Figure 3.** Digital-microscope images of through-thickness-edge-sectional view for the cross-ply-CFRP samples with thin and thick plies: (**a**) NA; (**b**) NB; (**c**) NC: (**d**) WB; (**e**) WC.

### 3.2. Static Tensile Properties

For the cross-ply-CFRP specimens with thin and thick plies, the stress–strain response was linearly proportional to failure. The tensile modulus ($E_L$) and Poisson's ratio ($\nu_{LT}$) were computed using a least-squares method involving the tensile stress–strain and longitudinal-strain–transverse-strain curves. The average tensile modulus, Poisson's ratio, tensile strength ($\sigma_{L.ult.ave}$), and failure strain ($\varepsilon_{L.ult.ave}$) of the CFRP specimens are summarized in Table 1.

**Table 1.** Static and fatigue tensile properties of the CFRP specimens.

| | NA | NB | NC | WB | WC |
|---|---|---|---|---|---|
| | | | Static tensile properties | | |
| Tensile modulus | 93.9 | 93.0 | 93.7 | 93.4 | 93.8 |
| $E_{L.ave}$ (GPa) | (4.4) | (3.1) | (7.1) | (3.6) | (3.3) |
| Poisson's ratio | 0.042 | 0.040 | 0.043 | 0.041 | 0.043 |
| $\nu_{LT.ave}$ | (0.001) | (0.005) | (0.009) | (0.006) | (0.002) |
| Tensile strength | 1.730 | 1.719 | 1.645 | 1.658 | 1.584 |
| $\sigma_{L.ult.ave}$ (GPa) | (0.131) | (0.105) | (0.068) | (0.103) | (0.041) |
| Failure strain | 1.878 | 1.817 | 1.747 | 1.731 | 1.715 |
| $\varepsilon_{L.ult.ave}$ (%) | (0.068) | (0.053) | (0.080) | (0.054) | (0.064) |
| | | | Fatigue tensile properties | | |
| Intercept $a$ (GPa) | 2.143 | 1.873 | 1.776 | 1.844 | 1.781 |
| Slope $ab$ | −0.182 | −0.130 | −0.103 | −0.116 | −0.104 |

() indicate standard deviations.

The tensile strength and failure strain of the thinnest 90°-ply CFRP (NA) specimens were 5% (strength) and 7% (strain) higher than those of the thin 90° ply (NB and NC). For the identical 90°-ply-thickness specimens, the tensile strength and failure strain of the cross-ply-CFRP specimens with the thin ply (NB and NC) were slightly higher than those of the cross-ply-CFRP specimens with the thick ply (WB and WC). However, the tensile moduli and Poisson's ratios were comparable between all the cross-ply-CFRP specimens with thin and thick plies. The volume fractions of the 0° plies, 90° plies, and fibers (total) were the same in all the CFRP specimens. Therefore, the results of the tensile moduli and Poisson's ratio seem reasonable, and were estimated using a simple rule of mixtures.

### 3.3. Fatigue Tensile Properties

Figure 4 indicates the *S–N* curves for the cross-ply-CFRP specimens with thin plies (NA, NB, and NC). The *S–N* curves for the cross-ply-CFRP specimens with thick plies (WB and WC) are also presented in this figure. The horizontal lines show the static tensile strengths. For the lower-fatigue-cycle testing (<$10^5$ cycles), the trends in the fatigue response were comparable to the static properties of the thin and thick plies and stacking sequences. The fatigue response of the thinnest 90°-ply-CFRP specimens were higher than those of the other 90°-ply specimens during the lower-fatigue-cycle testing (<$10^5$ cycles). However, for the higher-fatigue-cycle testing (>$10^5$ cycles), the fatigue responses decreased with a decrease in the 90°-ply thickness. The fatigue properties (slope and fatigue-limit stress, etc.) of the NA specimen were lower than those of the other cross-ply-CFRP specimens with thin plies (NB, NC) and of the cross-ply-CFRP specimens with thick plies (WB and WC).

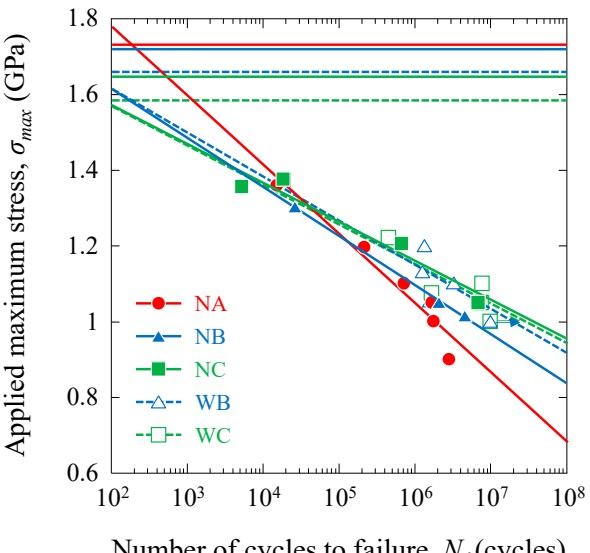

**Figure 4.** Relationship between the applied maximum stress and the number of cycles to failure, defined as the *S–N* curves for the cross-ply CFRPs with thin and thick plies.

## 4. Discussion

### 4.1. Static Tensile Properties

Figure 5 shows the digital-microscope images of the NA, NB, and NC specimens obtained from the in situ static tensile tests just before fracture. Transverse matrix cracking in relation to the thickness of the ply, delamination damage, and fiber breakage were observed in the specimens.

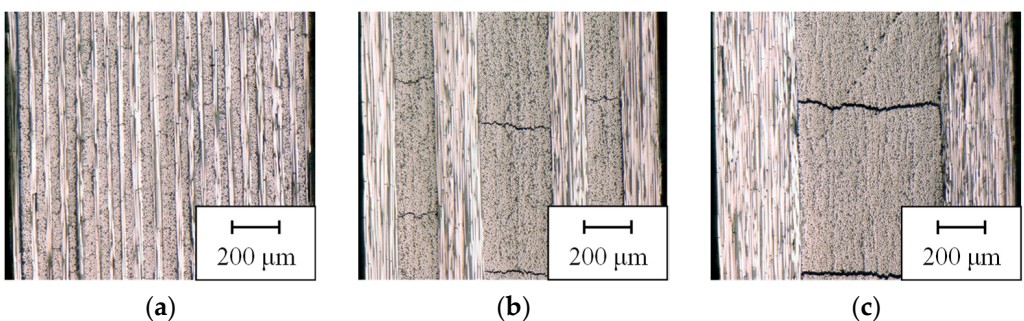

(a)          (b)          (c)

**Figure 5.** Digital-microscope images of the NA, NB, and NC specimens obtained from the in situ static tensile tests just before fracture: (**a**) NA at 1.65 GPa; (**b**) NB at 1.55 GPa; (**c**) NC at 1.45 GPa.

Figure 6 shows the relationship between the transverse-crack-onset stress and the stress first noticed in the transverse crack on an in situ static test. The 90°-ply thickness

or the thickness of the center ply, including the transverse-crack density, i.e., the number of cracks in a layer per unit length and the applied stress, are shown in Figure 6. The transverse-crack-onset stress increased with decreases in the 90°-ply thickness. The onset stress of the NA specimen (900 MPa) was more than twice as high as that of the NC specimen (400 MPa; 750 MPa in the NB specimen).

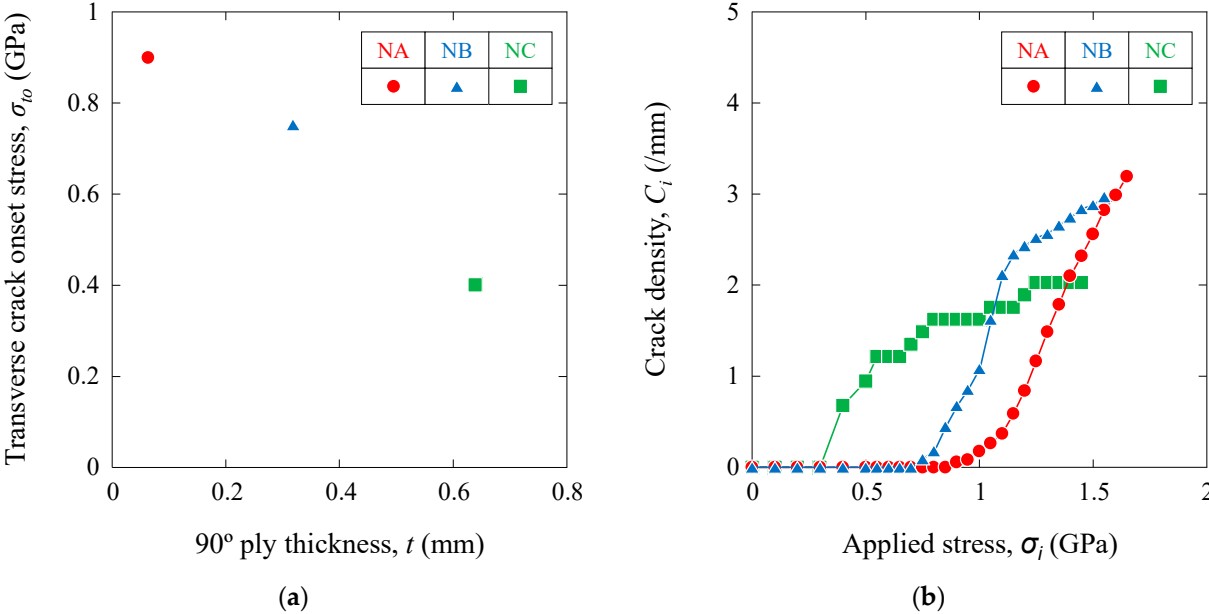

**Figure 6.** Relationship between (**a**) the transverse crack density and the applied stress and (**b**) the transverse-crack-onset stress and the 90°-ply thickness.

In this study, the transverse-crack-onset stress or strain was computed using a finite-element-method analysis with a Drucker–Prager elastoplastic damage model in the matrix, the cohesive-zone model in the fiber matrix, and 0°–90°-layer interfaces. The results showed that the transverse-crack-onset strain of the thin 90°ply was higher than that of the thick 90° ply. A similar finite-element method (FEM) analysis was performed in this study, and the results are shown in the Supplementary Material (Figures S1 and S2, and Table S1) [34,35]. Comparable findings have been observed experimentally and analytically in the literature [34].

The crack density increased quickly in the initial stage and then reached a saturation value. The critical crack density also increased with decreases in the 90°-ply thickness. For the NA (the thinnest 90° ply) specimens, the constraint effects of the 0° layers were large. The onset of the transverse matrix cracking in relation to the thickness of the ply, the delamination damage, and the fiber breakage was suppressed. Therefore, the tensile strength and failure strain of the NA specimen were higher than those of thel other cross-ply specimens.

For the identical 90°-ply-thickness specimens, the differences between the thin and thick ply specimens are thought to have been caused by the matrix layer, where the transverse crack propagation was slightly delayed. Therefore, the tensile strength and failure strain of the cross-ply-CFRP specimens with thin plies (NB and NC) were slightly higher than those of the cross-ply-CFRP specimens with thick plies (WB and WC).

*4.2. Fatigue Tensile Properties*

4.2.1. S–N Curves

The *S–N* curves can be described using a semi-log model. The semi-log model [36–41] is given by

$$\sigma_{max} = a + b \log\left(N_f\right) \qquad (4)$$

where $a$ and $b$ are the experimental constants. The least-squares fitting of the fatigue trends with the semi-log model is demonstrated in Figure 4. The intercept, $a$, and slope, $b$, are summarized in Table 1. The intercept, $a$, and the absolute value of the slope, $|b|$, increased with decreases in the 90°-ply thickness. The trend, $a$, depended on the static tensile strength. A higher $|b|$ indicates enormous fatigue degradation. Consequently, the fatigue characteristics of the NA specimens were lower than those of the NB, NC, and cross-ply-CFRP specimens with thick plies (WB and WC). The fatigue response of the thinnest 90°-ply-CFRP specimens were higher than those of the other 90°-ply-thickness specimens during the lower-fatigue-cycle testing ($<10^5$ cycles). The applied maximum stress of the thinnest 90°-ply-CFRP specimen was 3% higher than those of the other 90°-ply specimens at $10^4$ cycles, and the transverse-crack-onset stress under fatigue loading of the thinnest 90°-ply-CFRP specimen was 31% higher than that of the other 90°-ply specimens. However, the fatigue response of the thinnest 90°-ply-CFRP specimens were lower than those of the other cross-ply-CFRP specimens with thin plies and of the cross-ply-CFRP specimens with thick plies during the higher-fatigue-cycle testing ($>10^5$ cycles). The applied maximum stress of the thinnest 90°-ply-CFRP specimen was 8% lower than those of other 90°-ply specimens at $10^6$ cycles.

The fatigue response for the lower-fatigue-cycle testing ($<10^5$ cycles) of the cross-ply-CFRP specimens with thin plies (NB and NC) were slightly higher than those of the cross-ply-CFRP specimens with thick plies (WB and WC), which was attributed to the delayed transverse crack propagation in the matrix layer. However, the number of transverse cracks in the cross-ply-CFRP specimens with thin plies (NB and NC) was slightly larger than those in the cross-ply-CFRP specimens with thick plies (WB and WC). Therefore, the fatigue response during the higher-fatigue-cycle testing ($>10^5$ cycles) of the cross-ply-CFRP specimens with thin plies (NB and NC) were slightly lower than those of the cross-ply-CFRP specimens with thick plies (WB and WC).

### 4.2.2. Transverse-Crack-Onset Stress under Fatigue Loading

Figure 7 displays the relationship between the transverse-crack-onset stress under fatigue loading (first observed in the transverse crack on the in situ fatigue test using method (1), as described in Section 2.5) and the 90°-ply thickness (central 90°-ply specimen).

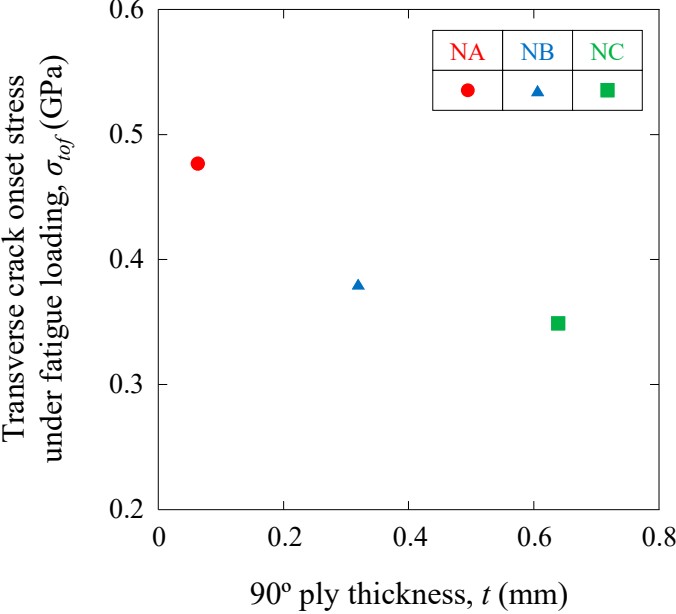

**Figure 7.** Relationship between the transverse-crack-onset stress under fatigue loading and the 90°-ply thickness.

The transverse-crack-onset stress under fatigue loading increased with the decrease in the 90°-ply thickness, although the onset stress was lower than in the static case. The onset stress of the NA specimen (477 MPa) was higher than that of the NC specimen (349 MPa; 381 MPa in the NB specimen).

### 4.2.3. Fatigue-Damage-Propagation Behavior

Figure 8 presents the digital-microscope images of the NA, NB, and NC specimens obtained from the in situ fatigue tensile test using method (2), as described in Section 2.5 at $N = 990,000$ cycles. The transverse matrix cracking in relation to the thickness of the ply, delamination damage, and fiber breakage increased with the number of cycles for all the specimens.

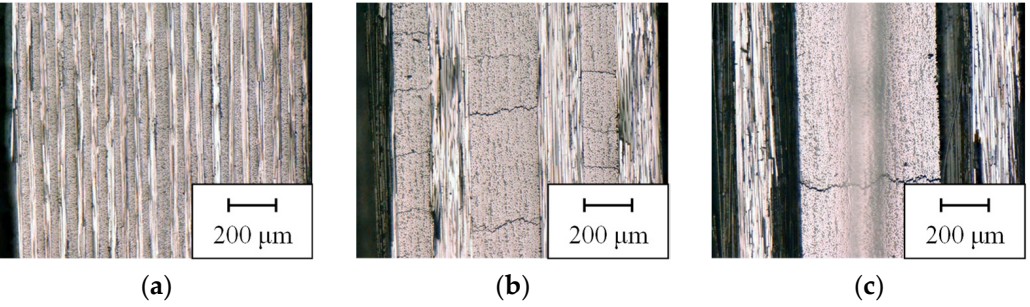

| (a) | (b) | (c) |

**Figure 8.** Digital-microscope images of the NA, NB, and NC specimens obtained from the in situ fatigue tensile tests at several $N = 990,000$ cycles . . . (**a**) NA; (**b**) NB; (**c**) NC.

Figure 9 displays the relationship between the transverse crack density, i.e., the number of cracks for a layer per unit length, and the number of cycles (in situ fatigue test using method (2), as described in Section 2.5). The crack density increased in the initial number of cycles and then reached a saturation value. The saturated crack densities that were similar to the static case and the saturated number of cycles increased with the decrease in the 90°-ply thickness. The fracture (i.e., transverse matrix cracking) of the low-elongation 90°-ply layers was delayed by the greater ductility of the high-elongation 0°-ply layers. Similar results were observed for unidirectional hybrid composites [4,5,29,30]. Depending on the distance between the 0°-ply layers, if the 90°-ply layer becomes thinner, the 0°-ply layer can carry the load until more transverse cracks occur.

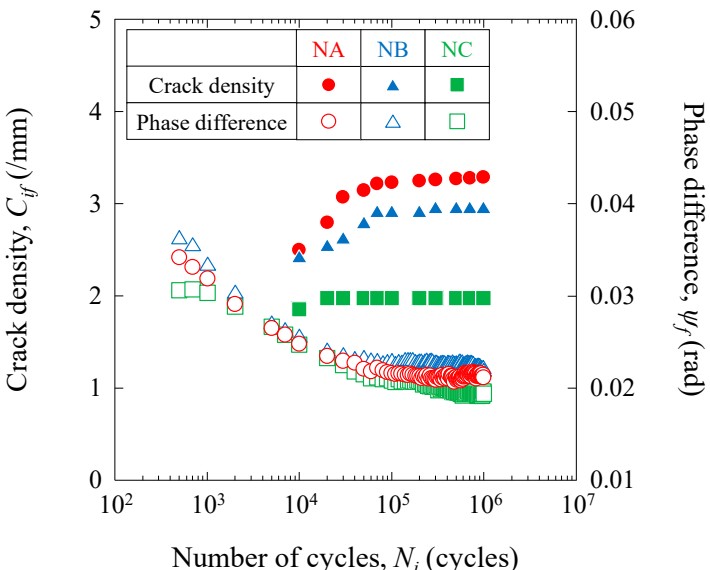

**Figure 9.** The transverse-crack density and the phase difference as a function of the number of cycles.

The figure also includes the relationship between the number of cycles and the phase variations between the load and the displacement in these cycles. The phase difference decreased in the initial number of cycles and then reached a saturation value (0.02 rad). The phase differences were caused by the viscoelastic nature of the 90° ply and changed with the 90° ply's transverse cracking and the 0° ply's saturated behavior. The 90° ply's transverse cracks increased with the increase in the number of cycles and the decreasing phase differences. The saturated-phase differences were comparable among the NA, NB, and NC specimens, and the number of saturated cycles increased with the decrease in the 90°-ply thickness.

Fatigue damage, such as matrix cracking and delamination, often results in a significant reduction in the moduli of composite laminates. Hence, it is crucial to develop an analytical model to describe the cumulative damage of composites due to fatigue based on apparent stiffness reduction [36–41]. The stiffness reduction reflected the damaged state under the different fatigue cycles after the distribution of the damage to the NA, NB, and NC specimens. The cumulative fatigue damage [5,36–42] for the NA, NB, and NC specimens, $D_i$, is defined as

$$D_i = 1 - \frac{E_i}{E_0} \tag{5}$$

where $E_0$ and $E_i$ represent the apparent stiffness at the first cycle and the *i*-th cycle, respectively.

Figure 10 shows the cumulative fatigue damage for the NA, NB, and NC specimens as a function of the number of cycles, $N_i$.

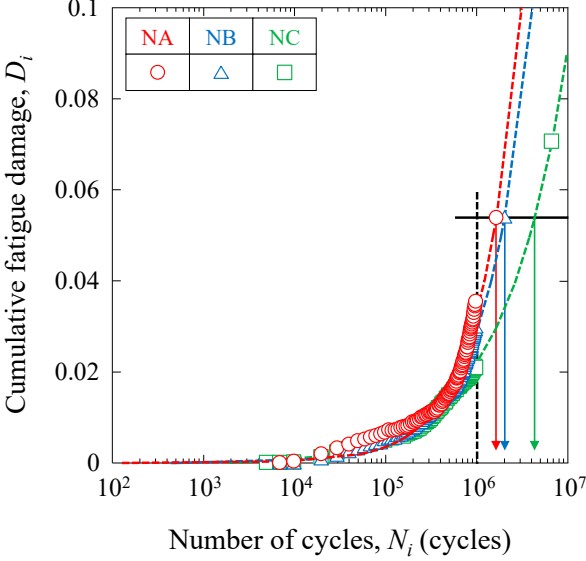

**Figure 10.** Cumulative fatigue damage as a function of the number of cycles.

The cumulative fatigue damage, $D_i$, for the NA, NB, and NC specimens increased with the increase in the $N_i$, and there was a relationship between $D_i$ and $N_i$, given by

$$\frac{N_i}{N_f} = C(D_i)^n \left\{ \frac{1 - \left( \frac{e^{D_{th}}}{e^{D_i}} \right)^{m_1}}{1 - \left( \frac{e^{D_i}}{e^{D_C}} \right)^{m_2}} \right\} \tag{6}$$

where $C$, $n$, $m_1$, and $m_2$ are the experimental constants. The $D_{th}$ and $D_C$ are the threshold and critical cumulative fatigue damages, respectively, and are assumed to be $D_{th} = 0$ and $D_C = 1$. The estimated relationship between $D_i$ and $N_i$ is also shown in Figure 10. The experimental results showed a reasonable agreement with the estimated relation obtained from Equation (6). Here, the experimental results were found to agree well with the estimation lines. Therefore, Equation (6) is effective for determining fatigue properties.

The $D_i$ at $10^6$ cycles increased with the decrease in the 90°-ply thickness (NA > NB > NC), as shown by the black dotted line in Figure 10. Assuming that the specimens fractured at the same $D_i$ ($N_f$ = 1,686,035 and $D_i$ = 0.946 for the NA specimen), the $N_f$ of the NB and NC specimens would be longer ($N_f$ = 2,070,118 (actually $N_f$ = 2,075,474) for the NB specimen and $N_f$ = 4,358,204 (actually $N_f$ = 6,870,901) for the NC specimen), as shown by the black solid line in Figure 10. The above estimation is thought to be applicable to lower-fatigue-cycle testing and with general composite materials.

For the NA (the thinnest 90° ply) specimens, the onset of the transverse matrix cracking in relation to the thickness of the ply and the increasing transverse cracking were suppressed. However, the damage to the fibers started near the transverse cracks, the greater saturated crack density (number of transverse cracks and several positions of the transverse crack) caused significant damage to the fibers in the 0° ply, and the delamination damage and fiber breakage increased rapidly because of the fatigue loading on the thin 0° ply. Therefore, the fatigue characteristics of the NA specimens were lower than those of the other cross-ply specimens.

The cumulative fatigue damage was thought to be the damage to the 0°-ply layers caused by the transverse cracks. The transverse cracks deflected into longitudinal cracks (i.e., delamination) near the 0°-ply layers and some cracks propagated into the 0°-ply layers. The greater the saturated crack density, the more fiber damage occurred. Therefore, the specimens with thinner 90°-ply layers and greater saturated crack density caused faster cumulative damage and shorter fatigue lifetimes. To control this phenomenon, some methods, such as the toughening of the matrix by modifying the nanostructure [43,44] and the insertion of ductile films between layers [45], have been proposed.

## 5. Conclusions

The static and fatigue tensile properties of cross-ply, high-strength polyacrylonitrile (PAN)-based carbon-fiber (T800SC)-reinforced epoxy-matrix composites (CFRPs) with thin plies were evaluated.

Under static loading, the tensile strength and failure strain of the thinnest 90°-ply-CFRP specimens were 5% (strength) and 7% (strain) higher than those of the other 90°-ply-thickness specimens. However, the tensile moduli and Poisson's ratios were similar across the cross-ply-CFRP specimens with thin and thick plies thickness. For the thinnest 90°-ply-CFRP specimen, the constraint effects of the 0° layers were large. The transverse-crack-onset stress of the thinnest 90°-ply-CFRP specimen was more than twice as high as that of the other 90°-ply specimens. The onset of transverse matrix cracking in relation to the thickness of the ply, delamination damage, and fiber breakage was suppressed.

Under fatigue loading, the trends in the fatigue response were similar to the static properties in the thin and thick plies and the stacking sequences during the lower-fatigue-cycle testing (<$10^5$ cycles). The fatigue responses of the thinnest 90°-ply-CFRP specimens were higher than those of the other 90°-ply-thickness specimens during the lower-fatigue-cycle testing (<$10^5$ cycles). The maximum applied stress of the thinnest 90°-ply-CFRP specimen was 3% higher than those of the other 90°-ply-thickness specimens at $10^4$ cycles, and the transverse-crack-onset stress under fatigue loading of the thinnest 90°-ply-CFRP specimen was 31% higher than that of the other 90°-ply specimens. The fatigue responses did, however, decrease with the reduction in the 90°-ply thickness during the higher-fatigue-cycle testing (>$10^5$ cycles), and the fatigue properties (the slope and fatigue-limit stress, etc.) of the thinnest 90°-ply-CFRP specimen were lower than those of the other cross-ply-CFRP specimens with thin plies and those of the cross-ply-CFRP specimens with thick plies. The maximum applied stress of the thinnest 90°-ply-CFRP specimen was 8% lower than those of the other 90°-ply-thickness specimens at $10^6$ cycles. For the thinnest 90°ply -CFRP specimens, the onset of transverse matrix cracking in relation to the thickness of the ply and the increasing transverse cracking were suppressed. However, the damage to the fibers began near the transverse cracks, and the 33% greater saturated crack density (the number of transverse cracks and their several positions) resulted in considerable damage

to the $0°$-ply fibers, and the delamination damage and fiber breakage increased rapidly because of the fatigue loading on the thin $0°$ ply.

The combination of transverse crack density under static and fatigue loading and cumulative-fatigue-damage evaluations were effective in clarifying the fractures in the CFRP laminates.

**Supplementary Materials:** The following supporting information can be downloaded at: https://www.mdpi.com/article/10.3390/jcs7040146/s1, Figure S1: (a) reference image, (b) representative volume element (RVE) of cross-ply with thin plies, and (c) RVE of cross-ply with thick plies; Figure S2: The n-plane principal-plastic-strain distributions of cross-ply CFRP laminates with thin and thick plies; Table S1: Material properties for FEM analysis.

**Author Contributions:** K.N.: Conceptualization, data curation, methodology, resources, supervision, validation, writing—reviewing and editing. Y.S.: data curation, methodology, visualization, software, formal analysis, investigation, validation, writing—original draft preparation. R.I.: resources, supervision, validation, writing—reviewing and editing. All authors have read and agreed to the published version of the manuscript.

**Funding:** This research was supported by Council for Science, Technology and Innovation (CSTI), cross-ministerial Strategic Innovation Promotion Program (SIP), "Materials Integration for revolutionary design system of structural materials" (Funding agency: Japan Science and Technology Agency (JST)).

**Institutional Review Board Statement:** Not applicable.

**Informed Consent Statement:** Not applicable.

**Data Availability Statement:** The datasets supporting the conclusions of this article are included within the article.

**Acknowledgments:** Not applicable.

**Conflicts of Interest:** The authors declare no conflict of interest.

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
