# Peer review of "Static and Fatigue Tensile Properties of Cross-Ply Carbon-Fiber-Reinforced Epoxy-Matrix-Composite Laminates with Thin Plies"

_jcs, doi:10.3390/jcs7040146_

Round 1
Reviewer 1 Report
The paper is noteworthy and presents very current engineering issues using current research techniques. The paper is an experimental and numerical study, where research based on the application of interdisciplinary research techniques is clearly exposed. In order for the paper to be published, it is necessary to make certain changes:
1. In the introduction, please clearly demonstrate the novelty of the present work in relation to other thematically similar research works.
2. in the introduction or a stage of the research description, please refer to works related to the study of thin-walled composite structures (DOI: 10.1016/j.compositesb.2021.109346) and to works in which similar research apparatus (DOI: 10.2478/ama-2023-0015) was used.
3. figure 5 should be enlarged a little.
4. in Figure 6 the vertical axis could be shown in the range of 0.2 to 0.6 because only in this range there are numerical values of the results.
5. The conclusions are described in terms of qualitative evaluation and not quantitative. Please describe them from a quantitative angle as well.
Author Response
Manuscript ID jcs-2308366
Dear Reviewer1,
Thank you very much for your email and the comments from the referee on our paper titled:
Manuscript ID jcs-2308366
“Static and fatigue tensile properties of cross-ply carbon fiber-reinforced epoxy matrix composite laminates with thin ply thickness”
by K. Naito, Y Seki, and R Inoue
We would like to revise our paper according to the referee's comments and your suggestions. Our responses are given below. We hope our responses and modification will be accepted for publication. We are looking forward to hearing from you. Thank you.
Very sincerely yours,
K. Naito

Reviewer 2 Report
Dear authors,
Hope this note find you well. You can find the reviewer's comments in the attached file.
Regards

Author Response
Manuscript ID jcs-2308366
Dear Reviewer2,
Thank you very much for your email and the comments from the referee on our paper titled:
Manuscript ID jcs-2308366
“Static and fatigue tensile properties of cross-ply carbon fiber-reinforced epoxy matrix composite laminates with thin ply thickness”
by K. Naito, Y Seki, and R Inoue
We would like to revise our paper according to the referee's comments and your suggestions. Our responses are given below. We hope our responses and modification will be accepted for publication. We are looking forward to hearing from you. Thank you.
Very sincerely yours,
K. Naito

Reviewer 3 Report
This paper investigates the static and tensile fatigue properties of cross-ply high-strength polyacrylonitrile (PAN)-based carbon (T800SC) fiber-reinforced epoxy matrix composites (CFRPs) with thin ply thickness. Paper is well written and prepared with nice figures. Some research results obtained are meaningful to promote the applications of cross-ply carbon fiber reinforced epoxy matrix composite laminates in structure engineering. However, the authors are encouraged to consider the following minor comments for necessary improvement.
1. Abstract:
(1) Please provide a brief background of current paper, and then present the improvement percentage of fatigue performance through the comparison.
(2) Replace “static; fatigue; tensile properties” with “static and fatigue properties” in the key word.
2. Introduction:
(1) The authors should enrich the introduction, including research background, research progress and the contribution and innovation of current papers.
(2) As known, CFRP is the ideal engineering material owing to the superior mechanical and durability performances. Relevant analysis and summary are important for readers to better understand this material. There are also lots of interesting papers on the mechanical properties and long-term performances of FRP. The authors are encouraged to make the supplements in the research background through reviewing the following relevant studies, such as Mechanics of Advanced Materials and Structures, 2023, 30(4):814-834, Composite Structures, 2021, 261: 113285, Engineering Structures, 2023, 274: 115176.
3. Materials and methods:
(1) Please provide the detailed structure diagram of laying-up for specimens.
(2) State the stress level applied in the fatigue test.
(3) For static and fatigue testing, how many specimens is repeated in one condition?What is standard for reference?
4. Results:
(1) Please provide the mechanism explanation for the tensile strength and failure strain of cross-ply CFRP with thin ply thickness (NB and NC) specimens and thick ply thickness (WB and WC).
(2) Check the mistake of sentence of “Figure 8 displays the relationship between the transverse crack density and the number of cycles ().”
(3) Can the function of cumulative fatigue damage and number of cycles be as the life prediction model for fatigue? Whether it can be extended to general composite materials?
5. Conclusion:
The conclusion is suggested to be further condensed according to the important findings, give the key results and highlight the innovation of this paper.
Author Response
Manuscript ID jcs-2308366
Dear Reviewer3,
Thank you very much for your email and the comments from the referee on our paper titled:
Manuscript ID jcs-2308366
“Static and fatigue tensile properties of cross-ply carbon fiber-reinforced epoxy matrix composite laminates with thin ply thickness”
by K. Naito, Y Seki, and R Inoue
We would like to revise our paper according to the referee's comments and your suggestions. Our responses are given below. We hope our responses and modification will be accepted for publication. We are looking forward to hearing from you. Thank you.
Very sincerely yours,
K. Naito

Author Response
Manuscript ID jcs-2308366
Dear Reviewer4,
Thank you very much for your email and the comments from the referee on our paper titled:
Manuscript ID jcs-2308366
“Static and fatigue tensile properties of cross-ply carbon fiber-reinforced epoxy matrix composite laminates with thin ply thickness”
by K. Naito, Y Seki, and R Inoue
We would like to revise our paper according to the referee's comments and your suggestions. Our responses are given below. We hope our responses and modification will be accepted for publication. We are looking forward to hearing from you. Thank you.
Very sincerely yours,
K. Naito

Round 2
Reviewer 1 Report
The paper can be published in its current form !
Reviewer 3 Report
Accepted
Reviewer 4 Report
Accept in present form